# Kinase Inhibitory Activities and Molecular Docking of a Novel Series of Anticancer Pyrazole Derivatives

**DOI:** 10.3390/molecules23123074

**Published:** 2018-11-24

**Authors:** Eman S. Nossier, Somaia S. Abd El-Karim, Nagy M. Khalifa, Ali S. El-Sayed, Emad S. I. Hassan, Salwa M. El-Hallouty

**Affiliations:** 1Pharmaceutical Chemistry Department, Faculty of Pharmacy, Al-Azhar University (Girls), Cairo 11754, Egypt; emy28_s@hotmail.com; 2Department of Therapeutical Chemistry, Pharmaceutical and Drug Industries Research Division, National Research Centre, Dokki, Giza 12622, Egypt; somaia_elkarim@hotmail.com (S.S.A.E.-K.); emadsayed98@yahoo.com (E.S.I.H.); 3Pharmaceutical Chemistry Department, Drug Exploration & Development Chair (DEDC), College of Pharmacy, King Saud University, Riyadh 11451, Saudi Arabia; 4Organic Chemistry Department, Faculty of Science, Al-Azhar University, Cairo 11651, Egypt; ali.sief1980@hotmail.com; 5Pharmacognosy Department, Drug Bioassay-Cell Culture Laboratory, National Research Center, Dokki, Giza 12622, Egypt; hallouty68@yahoo.com

**Keywords:** triarylpyrazole derivatives, triazolo[1,5-*a*]pyridines, anticancer activity, EGFR, molecular docking

## Abstract

A series of novel 1,3,4-triarylpyrazoles containing different heterocycles has been prepared, characterized and screened for their in vitro antiproliferative activity against HePG-2, MCF-7, PC-3, A-549 and HCT-116 cancer cell lines. The biological results revealed that compound **6** showed the highest anticancer activity so it was subjected to a kinase assay study where it reduced the activity of several protein kinases including AKT1, AKT2, BRAF V600E, EGFR, p38α and PDGFRβ at 100 μM using the radiometric or ADP-Glo assay method. Molecular docking simulation supported the initial kinase assay and suggested a common mode of interaction at the ATP-binding sites of these kinases, which demonstrates that compound **6** is a potential agent for cancer therapy deserving further research.

## 1. Introduction

Cancer is a main health issue in the world due to the yearly increases in the number of patient with this disease [1]. Alterations in cell cycle regulation may lead to the onset, progression and metastasis of cancer [2]. Protein kinases are involved in several biochemical mechanisms regulating the division, growth, and death of cells. The activation of these kinases in different cell signaling pathways has been implicated in cancer cell survival, invasiveness and drug resistance [3,4,5,6,7]. Unfortunately, the effectiveness of chemotherapy is limited by severe side effects, poor selectivity and drug resistance [8,9], so compounds targeting tyrosine kinases (EGFR, VEGFR and PDGFR) and serine/threonine kinases (B-RAF, AKT and p38) have become one of the most intensively pursued classes of cytotoxic agents [10,11]. 

Most antitumor agents possessing a pyrazole scaffold are reported to exert their action through inhibiting different enzymes (Figure 1). The approval of chemotherapeutics targeting inhibition of B-Raf kinase such as compound **A**, dual VEGFR and PDGFR kinases such as AZD2932, EGFR kinase such as compound **B**, AKT kinase such as AT7867 or p38 kinase such as SC-102 have been adopted to produce higher potency and selectivity [12,13,14,15,16].

Herein, we designed and prepared a novel group of 1,3,4-trisubstituted pyrazoles and examined their anticancer properties against HepG-2 (liver), MCF-7 (breast), PC-3 (prostate), A-549 (lungs) and HCT-116 (colon) cancer cell lines. Further kinase inhibition studies of the most potent compound were performed against twelve protein kinases [AKT1, AKT2, BRAF V600E, CDK2/CyclinA1, CHK1, EGFR, VEGFR-2, p38α, PDGFRβ, PI3K (p110a/p85a and p110b/p85a) and c-RAF] to know the exact mechanism of its anticancer activity. Additionally, molecular docking analyses were performed to explore the possible binding modes of the most active derivative against its biological targets hoping to facilitate the discovery of novel anticancer agents.

## 2. Results and Discussion

### 2.1. Chemistry

The reaction sequence for the synthesis of the target triarylpyrazoles **3**–**14** is outlined in Scheme 1 and Scheme 2. Reaction of the key precursor 2-((1,3-diphenyl-1*H*-pyrazol-4-yl)methylene)-malononitrile (**2**) with 2-cyanoacetohydrazide afforded the corresponding diaminopyridone derivative **3**. The IR spectra of product **3** displayed new strong bands at 3393, 3315 cm^−1^ assignable for two amino functions, besides another band at 1620 cm^−1^ for the 2-pyridone carbonyl group. The ^1^H-NMR spectrum revealed new two sharp singlets at 5.69 and 8.53 ppm related to the protons of two amino groups. Also, formation of compound **3** was confirmed on the basis of ^13^C-NMR with a signal appearing at 159.66 for the carbonyl carbon in the pyridine nucleus. Additionally, the MS of compound **3** revealed the presence of a molecular ion peak at *m*/*z* 393 corresponding to the molecular formula C_22_H_15_N_7_O. The target triazolo[1,5-*a*]pyridine derivatives **4a**–**g** were obtained through condensation of product **3** with a series of aromatic and aryl aldehydes (Scheme 1). The formation of the new products **4a**–**g** was confirmed by the disappearance of the NH_2_ group signals and the appearance of another due to a (-NH) function. The IR spectra of the title compounds exhibited bands in the 3294–3113 cm^−1^ range specific for -NH groups and 2221–2213 cm^−1^ due to CN functions besides strong bands at 1678–1653 cm^−1^ for (CO) functions. The ^1^H-NMR showed sharp singlets in the 8.46–8.31 ppm range due to (-NH) protons. In addition, ^13^C-NMR of the title compounds revealed the carbons at their expected regions and the molecular ion peaks in the MS confirmed the corresponding molecular formulas of the title products, please find more detailed data in the Appendix A.

Additionally, treatment of the key precursor **2** with barbituric acid in basic solution afforded the corresponding pyranopyrimidine derivative **5**. The IR spectrum showed absorptions at 3432, 3212, 2211 and 1741 cm^−1^ assignable to the amino, amide, cyano and carbonyl functions, respectively. Their ^1^H-NMR spectrum revealed four singlet signals specific for pyran ring protons, the amino function and 2 × NH protons at 4.67, 5.20, 9.79 and 11.21, respectively. MS proved the molecular weight of the expected structure, with the existence of a molecular ion peak [*M^+^*] at *m/z* = 424. Also, compound **2** was treated with series of cyclic ketones (1,3-indanedione, α-tetralone, or cyclohexanone) in presence of ammonium acetate to give the title products **6**–**8**. Compound **6** exhibited three bands at 3334, 2202 and 1708 cm^−1^ attributable to NH_2_, CN and CO functions, respectively, in its IR spectrum. In addition, the ^1^H-NMR spectum of compound **6** showed one sharp singlet at 6.89 ppm for the NH_2_ protons. The MS of **6** exhibited a molecular-ion peak at *m/z* 439.

The IR spectrum of product **7** showed strong bands at 3346, 2213 cm^−1^ indicating the existence of NH_2_ and CN groups. The ^1^H-NMR spectrum showed the presence of two peaks as triplets at 2.26 and 2.68 ppm for two CH_2_ moieties, and the NH_2_ protons appear as a singlet at δ 6.8 ppm. Also, the formation of compound **7** was confirmed by the existence of its molecular-ion peak at *m/z* 440. Moreover, compound **8** showed two strong bands at 3325 and 2217 cm^−1^ corresponding to NH_2_ and CN functions in its IR spectrum. The ^1^H-NMR spectra of **8** indicated multiplet peaks in the 1.55–2.77 ppm region equivalent to four methylene groups in the fused system and a singlet at δ 6.57 ppm of the NH_2_ group. The ^13^C-NMR spectrum of **8** displayed 22 distinct resonances which proved the suggested structure and a [M^+^]-ion peak at *m/z* 391 in the MS. On the other hand, the reaction of starting **2** with substituted hydrazines (hydrazine hydrate, phenyl hydrazine or methyl hydrazine) in refluxing ethyl alcohol containing piperidine caused the formation of pyrazole-3,5-diamine **9**, 5-imino-1-phenylpyrazol-3-amine **10** and 5-imino-1-methylpyrazol-3-amine **11** derivatives, respectively.

The IR spectrum of **9** showed absorptions bands at 3431, 3313 cm^−1^ corresponding to two amino functions, besides two singlet peaks were observed at 6.58 and the aromatic protons region due to amino and methine protons. Also, the MS gave a [M^+2^]-ion peak at *m/z* 328 equivalent to the molecular formula C_19_H_16_N_6_. Finally, treatment of compound **2** with different reagents such as 2-cyanoacetamide, urea or thiourea in the presence of sodium ethoxide afforded the corresponding 4,6-diamino-2-oxopyridine-3-carbonitrile derivative **12** and4,6-diamino-pyrimidin-2(5*H*)-one/thione derivatives **13a**,**b** (Scheme 2). The IR spectrum of **12** showed four bands at 3366, 3212, 2214 and 1690 corresponding to 2 × NH_2_, CN and C=O functions, respectively, whereas the two amino groups were observed as two singlet peaks in the 6.95 and 7.12 ppm region in addition to a sharp singlet peak appearing at 7.29 ppm belonging to methine proton in its ^1^H-NMR spectrum. The [M^+2^]-ion peak of 12 appeared at *m/z* 380, in agreement with the proposed structure. Also, derivatives **13a**,**b** were confirmed by the presence of strong bands around 3408–3162 cm^−1^ for 2 × NH_2_ functions, besides two absorption bands, one of them at 1725 cm^−1^ corresponding to the C=O of **13a** and the other observed at 1163 cm^−1^ in its IR spectrum, indicating the presence of the C=S function in **13b**. The ^1^H-NMR spectrum indicated the presence of methine protons at 7.04, 6.93 ppm as two singlets and the chemical shift of the 2 × NH_2_ protons in the range 8.46–9.26 ppm in the form of singlet peaks. Furthermore, the ^13^C-NMR and MS spectrum revealed the carbons at their expected regions and the molecular formulas of both compounds.

### 2.2. Biological Evaluation

#### 2.2.1. In Vitro Cytotoxic Screening

The nineteen new target compounds **3**–**13** were preliminary screened for their in vitro cytotoxic activity at a concentration of 100 μM against the HePG-2, MCF-7, PC-3, A-549 and HCT-116 cell lines following the formerly reported techniques [17,18] (Table 1).

Compounds **3**, **4a**, **4e**, **6**, **9**, **11**, **12a**, **12b**, **13a**, and **13b** that displayed cytotoxic activity higher than 80% at a concentration of 100 μM were used to calculate their IC_50_ values, which corresponds to the concentration required for 50% inhibition of cell viability. Doxorubicin, which is one of the most effective anticancer agents, was used as a reference drug (Table 2). Substitution at p-4 of 1,3,4-trisubstituted pyrazole moiety with diaminopyridone derivative gave **3** which has higher potency against HePG-2, PC-3 and A-549 cell lines (IC_50_ = 29.23, 18.81 and 33.07 μM) in comparison to doxorubicin (IC_50_ = 37.80, 41.10 and 48.80 μM), respectively, due to the presence of the two free amino groups attached to the pyridone scaffold. Cyclization of these two amino groups with different aldehydes afforded 2-substituted triazo[1,2,4]pyridone derivatives **4a**–**g**. When p-2 of the triazolopyridone moiety was substituted with a phenyl group as in **4a**, the cytotoxic activity was approximately retained or slightly decreased against HePG-2 (IC_50_ = 25.13 μM), decreased against PC-3 and A-549 cell lines (IC_50_ = 89.23, 108.20 μM) and greatly enhanced against MCF-7 (IC_50_ = 12.00 μM). Substitution with a five membered ring (compounds **4f**,**g**) or a phenyl having electron withdrawing or donating groups (compounds **4b**–**e**) drastically decreased or abolished the cytotoxic activity. Direct attachment of fused pyridine moieties at p-4 of the parent pyrazole in **7** and **8** decreased the cytotoxic activity against all tested cell lines except for 5-oxo-5*H*-indeno-[1,2-b]pyridine-3-carbonitrile (**6**), which revealed excellent potency against MCF-7, A-549 and HCT-116 cell lines (IC_50_ = 6.53, 26.40 and 59.84 μM), respectively. Attachment of a pyranopyrimidindione moiety at p-4 of the starting pyrzole scaffold in **5** resulted in moderate activity on PC-3 (IC_50_ = 55.61 μM) and a two fold decrease in the activity against the HCT-116 cell line (IC_50_ = 112.86 μM) in comparision with doxorubicin. Insertion at p-4 of the parent trisubstituted pyrazole, of a 3,5-diaminopyrazole moiety as in **9** or a 5-imino-1-phenyl-1*H*-pyrazole-3-amine as in **10** via a methylene linker led to drop in the potency against all cell lines, while substitution with 5-imino-1-methyl-1*H*-pyrazole-3-amine as in **11** elevated the activity against MCF-7 (IC_50_ = 18.22 μM). The 4,6-diaminopyridone derivative **12a** had excellent activity against HePG-2 and MCF-7 (IC_50_ = 39.96 and 36.67 μM) and moderate activity against PC-3 (IC_50_ = 78.34 μM), respectively. Double –O- replacement in diaminopyridone **12a** by an -S- as in diaminothiopyridine **12b** led to an increase in the potency against PC-3 (IC_50_ = 26.81 μM). Ring variation caused by replacing the pyridone moiety in **12a** with a thiopyrimidine one as in **13b** resulted in a noticeable decrease in the activity against all cancer cell lines. The oxopyrimidine derivative **13a** showed a remarkable increase in the potency, especially against HePG-2, MCF-7 and PC-3 (IC_50_ = 28.40, 48.49 and 67.06 μM), respectively. Finally, it could be concluded that the cytotoxic activity could be mainly attributed to the two free NH_2_ groups of the diaminopyridone moiety which was directly attached or through a methylene linker to p-4 of 1,3,4-trisubstituted pyrazole scaffold. Also, direct insertion of the 5-oxo-5*H*-indeno[1,2-b]pyridine system at p-4 resulted in excellent activity and compound **6** could be used as a lead compound for further preclinical studies in cancer treatment.

#### 2.2.2. Biochemical Assay (Kinase Inhibitor Activity)

Based on the *in-vitro* cytotoxicity screening results, the most potent compound **6** was selected for *in vitro* inhibition assessment versus a series of twelve protein kinases [AKT1, AKT2, BRAF V600E, CDK2/CyclinA1, CHK1, EGFR, VEGFR-2, p38α, PDGFRβ,PI3K (p110a/p85a and p110b/p85a) and c-RAF] at 100 μM using the radiometric or ADP-Glo assay method (KINEXUS Corporation, Vancouver, BC, Canada) [19,20]. Six of the selected kinases (AKT1, AKT2, BRAF V600E, EGFR, p38α, PDGFRβ) were strongly inhibited by more than 94% with the highest inhibition noted with EGFR at 99%. Four of the kinases (VEGFR-2, CDK2/Cyclin A1 and both of the PI3 kinases) gave moderate inhibitions ranging 47% to 76%. Only CHK1 showed a nominal inhibition at 8%. In contrast, compound **6** seemed to slightly activate the c-RAF kinase with an increase in counts of 43% over the control substrate values (Table 3). 

#### 2.2.3. Molecular Modeling Studies

Molecular docking studies were performed using the Molecular Operating Environment (MOE^®^) 2008.10 package [21] to gain a better understanding of the results obtained from the kinase inhibition assays (AKT1, AKT2, BRAF V600E, EGFR, p38α, PDGFRβ) and the target compound **6**. 

The three-dimensional X-ray structures of Akt1 (PDB: 4GV1) [22], AKT2 (PDB: 2JDR) [23], BRAF V600E (PDB: 3D4Q) [12], EGFR (PDB: 1M17) [24] and p38 alpha (PDB: 2EWA) [25] were used. The X-ray crystallography of the PDGFR β structure was not fully resolved [26].

As shown in Figure 2B, compound **6** occupied the ATP binding site of Akt1 kinase. In this binding model, Lys179 formed arene-cation interactions with the centroid of the pyrazole moiety, and a hydrogen bond with the inden[1,2-*b*]pyridine oxygen (distance: 2.92 Å). Besides, the hydrogen of NH_2_ attached to the indenopyridine scaffold served as an H-bond donor for the side chain of Glu234 (distance: 1.34 Å).

The binding model of compound **6** into AKT2 kinase is mediated by two hydrogen bonds as depicted in Figure 3. One H-bond appeared as a H-donor between a NH_2_ group hydrogen and the backbone of Leu158 (distance: 1.77 Å), and the other H-bond was linked with the side chain of Thr292 as a H-bond acceptor with indenopyridine oxygen (distance: 2.91 Å). Meanwhile, there were arene-cation interactions between the centroid of the phenyl ring at p-3 and Lys181, and arene-arene interactions between the centroids of the pyrazole and Phe163.

The binding of compound **6** into BRAF V600E (Figure 4) revealed three hydrogen bonds: two H-bonds were binding the two protons of the amino group, as H-donors with the backbones of Cys532 and Gly534 (distance: 1.65 and 1.46 Å), while the third H-bond was linking the nitrogen of the cyano group as a H-acceptor with the sidechain of Ser535 (distance: 2.51 Å). Moreover, the indeno[1,2-*b*]pyridine scaffold was inserted nicely inside the pocket via two arene-arene interactions, one between the centroid of the pyridine and Trp531, and the other between the phenyl ring and Phe583. The EGFR-binding domain in Figure 5 demonstrates that the nitrogen of the cyano group was involved in two H-bond acceptors with the sidechains of Arg817 and Asn818 (distance: 2.37, 2.99 Å, respectively).

Additionally, a H-bond donor interaction (Figure 6) was established between a proton of the NH_2_ group and the sidechain of Asp813 (distance: 1.45 Å). Furthermore, Lys721 shared the fixation of indeno[1,2-*b*]pyridinemoiety with the protein binding pocket via formation of two different interactions, an arene-cation interaction with the centroid of the phenyl ring and a H-bond acceptor with the carbonyl oxygen.

The above docking analysis was consistent with the kinase assay data. Furthermore, the results indicated that the introduction of indeno[1,2-*b*]pyridine scaffold to the pyrazole moiety at p-4 might reinforce the combination of compound **6** and receptors of AKT1, AKT2, BRAF V600E, EGFR and p38α, which might enhance the binding affinity, thus explaining the increased anticancer activity of this compound.

## 3. Experimental Section

### 3.1. General Information

Melting points (uncorrected) were determined on an Electrothermal 9100 apparatus (Cole- parmer, Staffordshire, United Kingdom). Elemental microanalyses were carried out using an Elementar system (Vario, Langenselbold, Germany) and the results were within the theoretical value ranges. Infrared spectra were recorded on a FT/IR-4100 instrument (Jasco, Kyoto, Japan), using KBr pellets. ^1^H-, ^13^C-NMR spectra were recorded on an AS-500 NMR spectrometer (JEOL, USA, Inc. CA, USA) or a Mercury Plus-Oxford 400 MHz (Palo Alto, California, USA) using TMS as an internal reference. The mass spectra were recorded on a GC MS-Qp1000EX system (Shimadzu corporation, Kyoto, Japan), and a MAT SSQ-7000 mass spectrometer (Finnigan, Pipersville, New Jersey, USA).

*1,6-Diamino-4-(1,3-diphenyl-1H-pyrazol-4-yl)-2-oxo-1,2-dihydropyridine-3,5-dicarbonitrile* (**3**): A mixture of freshly prepared 2-cyanohydrazide (0.02 mol) and compound **2** (0.01 mol) was refluxed for 1 h in ethyl alcohol (25 mL) containing a few drops of piperidine. The precipitated product was washed with ethanol and recrystallized from methyl alcohol. Yield: 72%; m.p.: 276–278 °C; IR (KBr, cm^−1^) *ν*: 3393, 3315 (2NH_2_), 2217 (CN), 1661 (CO); ^1^H-NMR (DMSO-*d*_6_): δ, 5.69 (s, 2H, NH_2_), 7.98–7.35 (m, 10H, ArH), 8.53 (s, 2H, NH_2_), 9.02 (s, 1H, CH); ^13^C-NMR (DMSO-*d*_6_): δ 159.66, 157.10, 152.40, 150.07, 139.28, 132.40, 130.28, 130.16, 129.26, 129.01, 127.59, 127.29, 118.89, 116.64, 115.70, 115.47, 88.03, 75.85; MS: [*m/z*, 393 (M^+^)]; Anal. Calcd. for C_22_H_15_N_7_O (393.40): C, 67.17; H, 3.84; N, 24.92; Found: C, 67.37; H, 3.48; N, 24.78.

#### 3.1.1. *7-(1,3-Diphenyl-1H-pyrazol-4-yl)-5-oxo-2-substituted-1,5-dihydro-[1,2,4]triazolo[1,5-a]pyridine-6,8-Dicarbonitriles*
**4a**–**g**

A equimolar amount of compound **3** (0.01 mol) and different aldehydes such as benzaldehyde, 4-fluorobenzaldehyde, 4-tolylaldehyde, 4-(dimethylamino)benzaldehyde, anisaldehyde, furan-2-carboxaldehyde, 5-methylfuran-2-carboxaldehyde or thiophene-2-carboxaldehyde) was refluxed for 6–8 h in of absolute ethyl alcohol (50 mL) containing a few drops of piperidine. The products formed were recrystallized from acetic acid.

*7-(1,3-Diphenyl-1H-pyrazol-4-yl)-5-oxo-2-phenyl-1,5-dihydro-[1,2,4]triazolo[1,5-a]pyridine-6,8-dicarbonitrile* (**4a**): Yield: 69%; m.p.: 252–255 °C; IR (KBr, cm^−1^) *ν*: 3290 (NH), 2216 (CN), 1670 (CO); ^1^H-NMR (DMSO-*d*_6_): δ 7.32–8.12 (m, 15H, ArH), 8.46 (s, 1H, NH), 9.00 (s, 1H, CH); ^13^C-NMR (DMSO-*d*_6_): δ 79.35, 88.03, 115.07, 115.46, 116.63, 116.82, 118.71, 118.92, 118.96, 119.00, 127.20, 127.28, 127.39, 129.24, 130.29, 138.15, 139.28, 150.06, 153.44, 153.94, 154.86, 157.10, 159.66, 173.36; MS: [*m/z* (%), 481 (M^+2^)]; Anal. Calcd. for C_29_H_17_N_7_O (479.49): C, 72.64; H, 3.57; N, 20.45; Found: C, 72.49; H, 3.41; N, 20.26.

*7-(1,3-Diphenyl-1H-pyrazol-4-yl)-2-(4-fluorophenyl)-5-oxo-1,5-dihydro-[1,2,4]triazolo[1,5-a]pyridine-6,8-dicarbonitrile* (**4b**): Yield: 66%; mp: 268–270 °C; IR (KBr, cm^−1^) ν: 3113 (NH), 2221 (CN), 1670 (CO);^1^H-NMR (DMSO-d_6_): δ 7.32–8.18 (m, 14H, ArH), 8.34 (br s, 1H, NH) 8.94 (s, 1H, CH); ^13^C-NMR (DMSO-d_6_): δ 82.14, 101.45, 115.10, 115.87, 119.79, 121.18, 123.68, 126.29, 126.84, 127.35, 128.56, 129.01, 129.31, 130.17, 132.94, 138.49, 150.16, 153.39, 157.24, 162.12, 165.01, 171.26; MS: [*m*/*z* (%), 499 (M^+2^)]; Anal. Calcd. for C_29_H_16_FN_7_O (497.48): C, 70.01; H, 3.24; N, 19.71; Found: C, 69.89; H, 3.12; N, 19.56.

*7-(1,3-Diphenyl-1H-pyrazol-4-yl)-5-oxo-2-(p-tolyl)-1,5-dihydro-[1,2,4]triazolo[1,5-a]pyridine-6,8-dicarbonitrile* (**4c**): Yield: 71%; m.p.: 276–278 °C; IR (KBr, cm^−1^) *ν*: 3288 (NH), 2217 (CN), 1678 (CO); ^1^H-NMR (DMSO-*d*_6_): δ 2.40 (s, 3H, CH_3_), 7.25–8.07 (m, 14H, ArH), 8.44 (br s, 1H, NH), 9.00 (s, 1H, CH); ^13^C-NMR (DMSO-*d*_6_): δ 21.54, 79.08, 84.91, 115.84, 116.16, 116.42, 116.76, 116.93, 117.49, 118.48, 118.72, 126.88, 127.31, 128.72, 129.58, 129.78, 129.91, 130.24, 132.73, 139.49, 147.54, 150.23, 153.73, 156.41, 161.79, 162.83, 164.17; MS: [*m/z* (%), 495 (M^+2^)]; Anal. Calcd. for C_30_H_19_N_7_O (493.52): C, 73.01; H, 3.88; N, 19.87; Found: C, 72.86; H, 3.69; N, 19.70.

*2-(4-(N,N-Dimethylamino)phenyl)-7-(1,3-diphenyl-1H-pyrazol-4-yl)-5-oxo-1,5-dihydro-[1,2,4]-triazolo-[1,5-a]pyridine-6,8-di-carbonitrile* (**4d**): Yield: 77%; m.p.: 283–285 °C; IR (KBr, cm^−1^) ν: 3130 (NH), 2213 (CN), 1668 (CO); ^1^H-NMR (DMSO-*d*_6_): δ 3.05 (s, 6H, 2CH_3_), 6.80–7.99 (m, 14H, ArH), 8.31 (s, 1H, NH), 9.03 (s, 1H, CH); ^13^C-NMR (DMSO-*d*_6_): δ 41.25, 84.06, 98.27, 114.34, 115.36, 116.48, 118.01, 121.10, 121.26, 126.15, 126.90, 127.42, 128.63, 129.11, 129.55, 130.10, 132.86, 140.02, 149.73, 150.45, 154.16, 164.20, 170.79; MS: [*m*/*z* (%), 524 (M^+2^)]; Anal. Calcd. for C31H22N8O (522.56): C, 71.25; H, 4.24; N, 21.44; Found: C, 71.08; H, 4.35; N, 21.26.

*7-(1,3-Diphenyl-1H-pyrazol-4-yl)-2-(4-methoxyphenyl)-5-oxo-1,5-dihydro[1,2,4]-triazolo-[1,5-a]pyridine-6,8-dicarbonitrile* (**4e**): Yield: 69%; m.p.: >300 °C; IR (KBr, cm^−1^) *ν*: 3283 (NH), 2213 (CN), 1653 (CO); ^1^H-NMR (DMSO-*d*_6_): δ 3.86 (s, 3H, OCH_3_),7.02–8.40 (m, 14H, ArH), 8.44 (s, 1H, NH), 9.02 (s, 1H, CH); ^13^C-NMR (DMSO-*d*_6_): δ 56.24, 76.38, 88.98, 114.52, 114.99, 116.36, 116.88, 118.88, 118.99, 124.74, 127.40, 127.63, 129.02, 129.55, 130.10, 132.29, 132.42, 139.21, 149.92, 152.62, 154.24, 156.91, 160.13, 164.14, 164.48, 172.25; MS: [*m/z* (%), 511 (M^+2^)]; Anal. Calcd. for C_30_H_19_N_7_O (509.52): C, 70.72; H, 3.76; N, 19.24; Found: C, 70.54; H, 3.59; N, 19.08.

*7-(1,3-Diphenyl-1H-pyrazol-4-yl)-2-(furan-2-yl)-5-oxo-1,5-dihydro-[1,2,4]-triazolo[1,5-a]pyridine-6,8-dicarbonitrile* (**4f**): Yield: 67%; m.p.: 246–248 °C; IR (KBr, cm^−1^) *ν*: 3284 (NH), 2216 (CN), 1663 (CO); ^1^H-NMR (DMSO-*d*_6_): δ 6.81 (m, 1H, CH), 7.38–7.99 (m, 13H, ArH), 8.46 (s, 1H, NH), 9.02 (s, 1H, CH); ^13^C-NMR (DMSO-*d*_6_): δ 81.31, 101.34, 110.46, 115.82, 119.89, 120.95, 126.34, 127.54, 128.59, 129.10, 129.35, 130.04, 131.81, 138.23, 141.87, 143.08, 150.19, 153.41, 159.37, 162.12, 169.33; MS: [*m/z* (%), 471 (M^+2^)]; Anal. Calcd. for C_27_H_15_N_7_O_2_ (469.45): C, 69.08; H, 3.22; N, 20.89; Found: C, 68.94; H, 3.09; N, 20.71.

*7-(1,3-Diphenyl-1H-pyrazol-4-yl)-5-oxo-2-(thiophen-2-yl)-1,5-dihydro-[1,2,4]-triazolo[1,5-a]-pyridine-6,8-dicarbonitrile* (**4g**): Yield: 65%; m.p.: 215–217 °C; IR (KBr, cm^−1^) *ν*: 3294 (NH), 2214 (CN), 1659 (CO); ^1^H-NMR (DMSO-*d*_6_): δ 7.27–8.06 (m, 13H, ArH), 8.40 (s, 1H, NH), 9.03 (s, 1H, CH); ^13^C-NMR (DMSO-*d*_6_): δ 76.39, 88.99, 115.02, 115.37, 115.55, 116.36, 116.70, 119.02, 127.26, 128.79, 129.01, 129.30, 130.32, 130.59, 132.27, 134.81, 136.30, 137.61, 139.29, 139.49, 141.88, 150.22, 152.43, 154.47, 156.94, 167.89; MS: [*m/z* (%), 487 (M^+2^)]; Anal. Calcd. for C_27_H_15_N_7_OS (485.52): C, 66.79; H, 3.11; N, 20.19; S, 6.60; Found: C, 66.51; H, 3.20; N, 20.28; S, 6.34.

#### 3.1.2. *7-Amino-5-(1,3-diphenyl-1H-pyrazol-4-yl)-2,4-dioxo-2,3,4,5-tetrahydro-1H-pyrano-[2,3-d]pyrimidine-6-carbonitrile* (**5**)

An equimolar amount of compound **2** and barbituric acid (0.01 mol) was heated for 2 h in absolute ethyl alcohol (40 mL) containing a few drops of piperidine. The product formed was washed with ethanol and recrystallized from ethanol. Yield 63%; m.p.: 187–189 °C; IR (KBr, cm^−1^) *ν*: 3432, 3212 (NH_2_, NH), 2211 (CN), 1741 (CO); ^1^H-NMR (DMSO-*d*_6_): δ 4.67 (s, 1H, CH), 5.20 (s, 2H, NH_2_), 7.44–7.93 (m, 10H, ArH), 8.19 (s, 1H, CH), 9.79, 11.21 (2s, 2H, 2NH); MS: *m/z* (%): 424 [M^+^, 2]; Anal. Calcd. for C_23_H_16_N_6_O_3_ (424.41): C, 65.09; H, 3.80; N, 19.80; Found: C, 65.23; H, 3.65; N, 19.59.

#### 3.1.3. *2-Amino-4-(1,3-diphenyl-1H-pyrazol-4-yl)-5-oxo-5H-indeno[1,2-b]pyridine-3-carbonitrile* (**6**), *3-amino-1-(1,3-diphenyl-1H-pyrazol-4-yl)-5,6-dihydrobenzo[f] quinoline-2-carbonitrile* (**7**) and *2-amino-4-(1,3-diphenyl-1H-pyrazol-4-yl)-5,6,7,8-tetrahydroquinoline-3-carbonitrile* (**8**)

An equimolar mixture of compound **2** (0.01 mol), and a cyclic ketone derivative, namely, 1,3-indanedione, α-tetralone, or cyclohexanone was heated under reflux for 3–5 h in absolute ethyl alcohol (40 mL), in the presence of ammonium acetate while reaction progress was monitored by TLC. The precipitated product precipitate was washed with ethanol and recrystallized from ethanol.

*2-Amino-4-(1,3-diphenyl-1H-pyrazol-4-yl)-5-oxo-5H-indeno[1,2-b]pyridine-3-carbonitrile* (**6**): Yield 68%, m.p. 209–211 °C; IR, ν: 3334 (NH_2_), 2202 (C≡N), 1708 (CO); ^1^H-NMR (DMSO-*d*_6_): δ 6.89 (s, 2H, NH_2_), 7.07–8.25 (m, 14H, ArH), 8.80 (s, 1H, CH); ^13^C-NMR (DMSO-*d*_6_): δ 89.21, 105.46, 117.09, 119.34, 119.89, 126.19, 126.98, 127.51, 127.82, 128.72, 129.14, 129.36, 130.22, 131.06, 132.88, 133.12, 134.61, 138.43, 143.01, 146.28, 153.37, 165.76, 167.91, 189.10; MS: *m/z* (%): 439 [M^+^, 3]; Anal. Calcd. for C_28_H_17_N_5_O (439.47): C, 76.52; H, 3.90; N, 15.94; Found: C, 76.38; H, 3.72; N, 15.75.

*3-Amino-1-(1,3-diphenyl-1H-pyrazol-4-yl)-5,6-dihydrobenzo[f]quinoline-2-carbonitrile* (**7**): Yield 69%, m.p. 223–225 °C; IR, ν: 3346 (NH_2_), 2213 (C≡N); ^1^H-NMR (DMSO-*d*_6_): δ 2.26 (t, 2H, CH_2_), 2.68, 6.81 (2t, 4H, 2CH_2_), 7.19–7.99 (m, 14H, ArH), 8.88 (s, 1H, CH); ^13^C-NMR (DMSO-*d*_6_): δ 26.79, 31.12, 87.49, 104.58, 116.97, 119.63, 119.90, 126.10, 126.32, 127.18, 127.45, 128.34, 128.68, 128.83, 129.20, 129.41, 131.23, 132.29, 133.07, 135.86, 138.65, 143.01, 146.27, 147.15, 159.45, 161.50; MS: *m/z* (%): 440 [M^+1^, 30]; Anal. Calcd. for C_29_H_21_N_5_ (439.51): C, 79.25; H, 4.82; N, 15.93; Found: C, 79.55; H, 4.61; N, 15.78.

*2-Amino-4-(1,3-diphenyl-1H-pyrazol-4-yl)-5,6,7,8-tetrahydroquinoline-3-carbonitrile* (**8**): Yield 74%, m.p. 269–271 °C; IR, ν: 3325 (NH_2_), 2217 (C≡N); ^1^H-NMR (DMSO-*d*_6_): δ 1.55–2.08 (m, 4H, 2CH_2_), 2.26–2.77 (m, 4H, 2CH_2_), 6.57 (s, 2H, NH_2_), 7.30–7.94 (m, 10H, ArH), 8.76 (s, 1H, CH); ^13^C-NMR (DMSO-*d*_6_): δ 21.54, 22.61, 26.12, 33.26, 89.64, 116.50, 117.09, 118.66, 120.14, 126.65, 127.20, 128.81, 129.22, 129.34, 130.17, 132.93, 139.53, 146.58, 149.55, 158.53, 161.77, 172.55; MS: *m/z* (%): 391 [M^+^, 100]; Anal. Calcd. for C_25_H_21_N_5_ (391.47): C, 76.70; H, 5.41; N, 17.89; Found: C, 76.45; H, 5.23; N, 17.68.

#### 3.1.4. *4-((1,3-Diphenyl-1H-pyrazol-4-yl)methylene)-4H-pyrazole-3,5-diamine* (**9**), *4-((1,3-diphenyl-1H-pyrazol-4-yl)methylene)-5-imino-1-phenyl-4,5-dihydro-1H-pyrazol-3-amine* (**10**) and *4-((1,3-diphenyl-1H-pyrazol-4-yl)methylene)-5-imino-1-methyl-4,5-dihydro-1H-pyrazol-3-amine* (**11**)

An equimolar mixture of compound **2** (0.01 mol) and a hydrazine derivative (hydrazine hydrate, phenyl hydrazine or methyl hydrazine) was refluxed for 6 h in absolute ethanol (50 mL) containing a few drops of piperidine. The solid precipitate produced was collected and recrystallized from methanol.

*4-((1,3-Diphenyl-1H-pyrazol-4-yl)methylene)-4H-pyrazole-3,5-diamine* (**9**): Yield 73%, m.p. 134–135 °C; IR, ν: 3431, 3313 (2NH_2_); ^1^H-NMR (DMSO-*d*_6_): δ 6.58 (s, 4H, 2NH_2_), 7.30–8.00 (m, 11H, ArH + =CH), 8.65 (s, 1H, CH); ^13^C-NMR (DMSO-*d*_6_): δ 112.81, 118.85, 119.31, 125.93, 126.91, 127.19, 128.39, 128.58, 129.68, 128.93, 129.07, 129.26, 130.02, 131.48, 132.65, 133.24, 139.75, 150.36, 151.87, 163.89; MS: *m/z* (%): 328 [M^+2^, 2]; Anal. Calcd. for C_19_H_16_N_6_ (328.37): C, 69.50; H, 4.91; N, 25.59; Found: C, 69.82; H, 4.69; N, 25.72.

*4-((1,3-Diphenyl-1H-pyrazol-4-yl)methylene)-5-imino-1-phenyl-4,5-dihydro-1H-pyrazol-3-amine* (**10**): Yield 77%, m.p. 206–208 °C; ^1^H-NMR (DMSO-*d*_6_): δ 6.75 (br s, 2H, NH_2_), 7.03–8.01 (m, 16H, ArH+ =CH), 8.89 (br s, 1H, NH), 9.09 (s, 1H, CH); ^13^C-NMR (DMSO-*d*_6_): δ 104.79, 115.85, 118.47, 119.94, 123.86, 126.14, 127.36, 128.75, 129.18, 129.39, 129.55, 130.12, 133.08, 138.11, 140.21, 145.96, 151.27, 153.01, 163.87; Anal. Calcd. for C_25_H_20_N_6_(404.47): C, 74.24; H, 4.98; N, 20.78; Found: C, 74.05; H, 4.82; N, 20.53.

*4-((1,3-Diphenyl-1H-pyrazol-4-yl)methylene)-5-imino-1-methyl-4,5-dihydro-1H-pyrazol-3-amine* (**11**): Yield 63%, m.p. 173–175 °C; IR, ν: 3401, 3214 (NH_2_, NH); ^1^H-NMR (DMSO-*d*_6_): δ 2.78 (s, 3H, CH_3_), 6.62 (s, 2H, NH_2_), 7.32–8.12 (m, 11H, ArH+ =CH), 8.68 (s, 1H, CH), 9.31 (s, H, NH); ^13^C-NMR (DMSO-*d*_6_): δ 26.71, 118.80, 119.18, 119.84, 126.72, 128.36, 128.68, 129.05, 129.32, 131.53, 131.95, 136.27, 137.25, 139.19, 152.67, 163.87; MS: *m/z* (%): 341 [M^−1^, 10]; Anal. Calcd. for C_20_H_18_N_6_ (342.4): C, 70.16; H, 5.30; N, 24.54; Found: C, 69.98; H, 5.14; N, 24.39.

#### 3.1.5. *4,6-Diamino-5-((1,3-diphenyl-1H-pyrazol-4-yl)methylene)-2-oxo-2,5-dihydropyridine-3-carbonitrile* (**12**) and *4,6-Diamino-5-((1,3-diphenyl-1H-pyrazol-4-yl)methylene)pyrimidin-2(5H)-one/thione* (**13a**,**b**)

An equimolar mixture of compound **2** (0.01 mol) and 2-cyanoacetamide, urea or thiourea was refluxed for 3–5 h in sodium ethoxide solution (sodium metal (0.01 mol) in 40 mL of absolute ethanol) while the reaction progress was under TLC control. The residue obtained upon pouring onto ice/water containing a few drops of hydrochloric acid (pH ~ 6) was recrystallized from ethanol.

*4,6-Diamino-5-((1,3-diphenyl-1H-pyrazol-4-yl)methylene)-2-oxo-2,5-dihydropyridine-3-carbonitrile* (**12**): Yield 73%, m.p. > 300 °C; IR, ν: 3366, 3212 (2NH_2_), 2214 (CN), 1690 (CO); ^1^H-NMR (DMSO-*d*_6_): δ 6.95, 7.12 (2s, 4H, 2NH_2_), 7.29 (s, 1H, CH), 7.33–7.96 (m, 10H, ArH), 8.66 (s, 1H, CH); ^13^C-NMR (DMSO-*d*_6_): δ 89.68, 104.59, 115.69, 119.88, 126.19, 127.49, 128.32, 129.10, 129.35, 129.68, 130.08, 131.26, 133.05, 138.91, 151.12, 165.45, 167.15, 184.99; MS: *m/z* (%): 380 [M^+^, 3]; Anal. Calcd. for C_22_H_16_N_6_O (380.40): C, 69.46; H, 4.24; N, 22.09; Found: C, 69.71; H, 4.38; N, 22.27.

*4,6-Diamino-5-((1,3-diphenyl-1H-pyrazol-4-yl)methylene)pyrimidin-2(5H)-one* (**13a**): Yield 67%, m.p. 163–165 °C; IR, ν: 3366, 3162 (2NH_2_), 1725 (C=O); ^1^H-NMR (DMSO-*d*_6_): δ 7.04 (s, 1H, CH), 7.21–7.94 (m, 10H, ArH), 8.86 (1s, 2H, NH_2_), 9.12 (s, 1H, CH), 9.26 (1s, 2H, NH_2_); ^13^C-NMR (DMSO-*d*_6_): δ 115.33, 117.18, 118.52, 118.82, 119.35, 127.10, 128.45, 129.26, 130.08, 130.92, 139.68, 151.57, 152.10, 160.15, 160.69, 174.37, 178.13, 178.94; MS: *m/z* (%): 356 [M^+^, 2]; Anal. Calcd. for C_20_H_16_N_6_O (356.38): C, 67.40; H, 4.53; N, 23.58; Found: C, 67.23; H, 4.35; N, 23.41.

*4,6-Diamino-5-((1,3-diphenyl-1H-pyrazol-4-yl)methylene)pyrimidine-2(5H)-thione* (**13b**): Yield 63%, m.p. 172–175 °C; IR, ν: 3408, 3341 (2NH_2_), 1163 (C=S); ^1^H-NMR (DMSO-*d*_6_): δ 6.93 (s, 1H, CH), 7.10–7.96 (m, 10H, Ar-H), 8.21 (s, 1H, CH), 8.46, 8.98 (2s, 4H, 2NH_2_); ^13^C-NMR (DMSO-*d*_6_): δ 114.68, 118.50, 118.88, 118.93, 119.07, 127.32, 127.82, 128.78, 129.05, 130.08, 130.31, 150.35, 161.56, 165.84, 172.97; MS: *m/z* (%): 371 [M^−1^, 60]; Anal. Calcd. for C_20_H_16_N_6_S (372.45): C, 64.50; H, 4.33; N, 22.56; Found: C, 64.31; H, 4.16; N, 22.39.

## 4. Conclusions

In summary, a series of 1,3,4-triarylpyrazole derivatives bearing different nitrogenous moieties were synthesized. Five human cancer cell lines (HePG-2, MCF-7, PC-3, A-549 and HCT-116) were utilized to estimate the cytotoxic properties of the obtained products. Compared to doxorubicin as a reference drug, six derivatives—**3**, **4a**, **6**, **12a**, **12b** and **13a**—were more potent against one or more cell lines. Target product **6** having the promising cytotoxic actions revealed excellent inhibitory activity against six kinases (AKT1, AKT2, BRAF V600E, EGFR, p38α and PDGFRβ) at 100 μM. Molecular modeling studies were done to validate the obtained pharmacological data and provide evidence for the observed anticancer behavior.

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
