# Peer review of "Kinase Inhibitory Activities and Molecular Docking of a Novel Series of Anticancer Pyrazole Derivatives"

_molecules, 2018, doi:10.3390/molecules23123074_

Reviewer 1 Report

I suggested may be revise introduction should be still more information about molecular docking studies in cancer.

Discussion should be more detail and discuss your current results and conclude with future directions, benefits these work in cancer patients drug therapeutics.

Author Response

Review Report Form  1

Comments and Suggestions for Authors:

Q: I suggested may be revise introduction should be still more information about molecular docking studies in cancer.

Changes were done (written in the manuscript)

Q: Discussion should be more detail and discuss your current results and conclude with future directions, benefits these work in cancer patients drug therapeutics.

Changes were done (written in the manuscript)

Reviewer 2 Report

The quality of all the figures must be improved.

The caption of figure 1 must describe in a more effective way the figure

Doxorubicin interacts with DNA by intercalation and a better positive control with a similar action of the studied compound must be used as a positive control for all of the experiments.

To effectively say that the most potent compound 6 is the most potent compound the authors must verify the IC50 of all the compounds in all of the studied cell lines. 

How the authors can explain the activation of c-Raf by compound 6? They should argue about that considering that this kinase can be overexpressed in some of the studied cell lines

A validation of the docking procedure is mandatory for all of the studied compounds.

All of the NMR, IR and mass spectra of the new compounds must be added in a supplementary material file.

Author Response

Review Report Form -2

Comments and Suggestions for Authors:

Q: The quality of all the figures must be improved.

Changes were done

The caption of figure 1 must describe in a more effective way the figure

Changes were done

Q: Doxorubicin interacts with DNA by intercalation and a better positive control with a similar action of the studied compound must be used as a positive control for all of the experiments.

First, we used doxorubicin as a potent cytotoxic drug to compare with our compounds. Then, we wanted to determine the mechanism of action of our compounds by using assay on multi and different enzymes at 100 µM.

In the future work, we will use reference drug (s) like sorafenib having the same mechanism of action of our compounds to compare the potency.

Q: To effectively say that the most potent compound 6 is the most potent compound the authors must verify the IC50 of all the compounds in all of the studied cell lines. 

All  new target compounds 3-13 were preliminary screened for their in vitro cytotoxic properties against HePG-2, MCF-7, PC-3, A-549 and HCT-116 cell lines with doxorubicin as a reference drug following the formerly reported techniques [17,18] at a concentration of 100 μM (Table 1) which showed the % inhibition. Compounds 3, 4a, 4e, 6, 9, 11, 12a, 12b, 13a, and 13b displayed cytotoxic action more than 80 % at a concentration of 100 μM were applied to obtain their IC50 rates. (Written in the text)

In another sentence, we calculate IC50 only for compounds having  % inhibition ≥ 80 on cancer cell lines at a concentration of 100 μM

If less than 80%, IC50 value of the compounds will be large and not accepted. So, we didn't calculate it and considered these compounds less potent.

Q: How the authors can explain the activation of c-Raf by compound 6? They should argue about that considering that this kinase can be overexpressed in some of the studied cell lines

we agree with you about that. So, in the future we decide to prepare a new series of 1,3,4-trisubstitued pyrazoles with different substitutions at p-4 plus our compounds in this  article and make kinase assay against c-raf to predict the SAR required to overcome this problem.

Q: A validation of the docking procedure is mandatory for all of the studied compounds.

- We did kinase assay for the most potent compound 6 against the cancer cell lines followed by docking of this compound against the targeted enzymes to explain the interaction between them.

- Compound 6 is active against six enzymes, so docking of all compounds against these enzymes will give large database.

In the future study, we will concentrate our work on EGFR enzyme (having the highest % inhibition) and make assay for all compounds at different concentration to calculate IC50's followed by docking for all compounds and compare the results.

All of the NMR, IR and mass spectra of the new compounds must be added in a supplementary material file.

Submission Date

Round  2

Reviewer 2 Report

The quality of all the figures was not improved, they are just larger. Moreover, the molecules are represented in different styles and these must be represented all in the same style. Figure 1 is also squashed.

Reviewer 1 suggestions should be better addressed.

The supplementary file with all of the NMR, IR and mass spectra of the new compounds is mandatory in my opinion and the paper cannot be accepted without it.

Author Response

Open Review

Q: The quality of all the figures was not improved, they are just larger.

We already increase the size of 2D figures because the MOE program can't give pictures larger and improved than that. But we changed the style of 3D figures to be improved.

Q: Moreover, the molecules are represented in different styles and these must be represented all in the same style.

We can control the direction in 3D picture to have the same style of the molecules with the most convenient look by rotation of the whole binding pocket enclosing our compound but we can't do that with 2D picture because the compound makes different conformations or poses during docking and we choose the most stable and relaxed conformer having the less binding energy.

Q: Figure 1 is also squashed.

We reinserted the figure in the text again.

Q: Reviewer 1 suggestions should be better addressed

We added the following sentences in the manuscript:

The activation of these kinases in different cell signaling pathways has been implicated in cancer cell survival, invasiveness and drug resistance [3-7]. (introduction, p.1)

Additionally, the molecular docking analyses were performed to explore the possible binding modes of the most active derivative against its biological targets hoping to facilitate the discovery of novel anticancer agents. (introduction, p.2)

We made changes in the text of Discussion of in vitro cytotoxic screening, p.6 & 7 highlighted with red color.

Depending on the results of in-vitro cytotoxic screening, the most potent compound 6 was selected for in vitro inhibition assessment versus a series of twelve protein kinases [AKT1, AKT2, BRAF V600E, CDK2/CyclinA1, CHK1, EGFR, VEGFR-2, p38α, PDGFRβ,PI3K (p110a/p85a and p110b/p85a) and c-RAF] at 100 μM using the radiometric or ADP-Glo assay method in KINEXUS Corporation, Vancouver, British Columbia, Canada [19,20]. (Discussion of biochemical assay, p.8)

Molecular modeling studies were done herein to validate the attained pharmacological data and give understandable evidence for the observed anticancer behavior. (conclusion, p.19)

Q: The supplementary file with all of the NMR, IR and mass spectra of the new compounds is mandatory in my opinion and the paper cannot be accepted without it.

Supplementary data has been uploaded.

Round  3

Reviewer 2 Report

the authors addressed most of the points suggested by the reviewer.